# Barriers to Health and Social Services for Unaccounted-For Female Migrant Workers and Their Undocumented Children with Precarious Status in Taiwan: An Exploratory Study of Stakeholder Perspectives

**DOI:** 10.3390/ijerph20020956

**Published:** 2023-01-05

**Authors:** Ming Sheng Wang, Ching-Hsuan Lin

**Affiliations:** 1Graduate Institute of Social Work, National Chengchi University, Taipei 116011, Taiwan; 2Department of Social Work, National Taiwan University, Taipei 10617, Taiwan

**Keywords:** unaccounted-for migrant workers, undocumented/stateless children, health, UNCRC

## Abstract

Unaccounted-for migrant workers (UMWs), who have left their employment placement and whose whereabouts are unknown, make up a vulnerable population in Taiwan. The children of UMWs have a particularly precarious status because they are undocumented/stateless, immigrant, and young. Living with this precarious status limits their children’s rights to survival and development. Moreover, services for female UMWs and their undocumented children are underdeveloped. This study explores the accessibility and availability of social services for UMWs and undocumented children, based on interviews with 12 stakeholders from multiple systems, including a local government, a child welfare placement center, a migrant worker detention center, a hospital, a regional religious center, and a foreign country office. Preliminary findings indicate the following: First, UMWs’ rights to healthcare are not preserved, and they experience greater prenatal risks because their illegal status excludes them from universal health coverage. Second, undocumented children’s rights to survival and development are concerning because these children can be placed in residential care without individualized care or environmental stimulation. Third, children’s rights to cultural identity and permanency are uncertain in that repatriation or adoption does not guarantee their future best interests.

## 1. Introduction

By January 2022, Taiwan had 664,733 legal transnational labor migrants acting as industrial workers or care workers, mostly from Southeast Asian countries, such as Indonesia, the Philippines, Thailand, and Vietnam [1]. There are another 50,000 unaccounted-for migrant workers (UMWs), who have left their placement of employment and whose whereabouts are unknown [2,3,4]. In other words, they illegally ran away from their employment and used to be referred to as “runaway workers” or “illegal immigrants.”

Some female UMWs give birth while or before going missing from their employment, and their illegal status makes their children undocumented or stateless. Currently, there are about 800 undocumented children in Taiwan [4,5,6]. The irregular immigration status of the mothers and children leads to multiple vulnerabilities: being a child or a woman, an immigrant, and an undocumented person [7,8].

Services for female migrant workers (MWs) and their undocumented children are underdeveloped in Taiwan. They have limited access to basic rights, such as healthcare and childcare, because the central government has not provided a complete policy response to these immigrants and their children’s precarious status. Additionally, very limited attention has been paid to this population in research. Following the principles of the United Nations Convention on the Rights of the Child (UNCRC) and the Universal Declaration of Human Rights (UDHR), nations should preserve the fundamental rights granted to every individual and provide appropriate assistance and protection in response to recognizing children’s identity, nationality, and family relations. This urges us to explore the needs of these UMWs and their children. Thus, this research, designed as an exploratory study, attempted to identify the precarious status of UMWs and their undocumented children and raise concerns regarding how the Taiwanese social service system responds to their needs.

### 1.1. Unaccounted-For Migrant Workers in Taiwan

MWs officially began entering Taiwan in 1989, and in the 1990s, the Taiwanese government offered work permits for MWs to legally work in labor-intensive industries (e.g., textiles and manufacturing industries) as industrial workers and in social welfare sectors (e.g., childcare, long-term care) as care workers [9]. There were 664,733 MWs as of the end of January 2022, representing a 220-times increase since 1989. With the increased number of MWs, social issues have been emerging, related to discrimination and labor exploitation against these MWs. Some of these issues conclude with the decision to leave their placement of employment and seek other job opportunities without a legal work permit. Reasons for their illegal leave include no freedom to transfer employment owing to the migrant worker policy, an imbalance between salary and workload, unexpected overtime hours or assignments, the expiry of the contract, conflicts with employers, or maltreatment by the employer or agent [9,10,11]. These workers were previously called “runaway migrant workers,” with stigmas associated with illegal residence and crimes. In 2017, the Taiwanese government referred to them as “unaccounted-for migrant workers” to destigmatize the image of being illegal and to promote their basic rights.

The lack of legal status leads to a variety of difficulties; many UMWs are more vulnerable and experience the danger of crime, exploitation, and poor healthcare [9]. These UMWs become not covered by national health insurance or labor insurance. When they are ill or injured, they are fearful of seeking help from formal systems and may receive only low-quality services from informal or illegal healthcare settings. Moreover, these UMWs are likely to search for jobs in the labor black market, where forced labor and exploitation may be more common [9]. Because UMWs are afraid of being reported and deported, they become more easily controlled by illegal agents. Thus, there are increased social issues resulting from their illegal, uncertain, and unexpected status. The vulnerability of these UMWs deserves much attention from the Taiwanese society and government, both for the benefit of the UMWs and to limit illegal activities that could happen.

### 1.2. Precarious Status of Unaccounted-For Migrant Workers and Undocumented Children

About half of the UMWs are women, who may become pregnant while or before being unaccounted [2,3,4]. For some workers, pregnancy is the reason for leaving their employment because they can be considered unqualified for labor and then be laid off or forced to return to their home country. For others, pregnancy while being missing can result in a more precarious situation for both the mothers and children. According to the Control Yuan [12] of the Taiwan government, an independent investigatory and auditory agency, many migrant worker mothers are unable to provide appropriate childcare and thus leave their children unaccompanied, stateless, or undocumented.

From January 2007 to July 2020, there were 941 unaccompanied children reported whose mothers were UMWs [5]. However, this number is based on reports from medical settings and could be substantially under-reported. Control Yuan commissioners have estimated that there was an increase of about 700 stateless or undocumented children each year, and there could be currently more than 20,000 unaccompanied children who were born unreported in private settings and thus without a birth certificate or citizenship [2,12].

The issue of stateless or undocumented children is a global one. In developed countries such as North America and Europe, millions of migrants are estimated to be undocumented, representing 10% to 15% of the migrant population [13,14]. The growing undocumented population has a large proportion of families with children who are likely to be undocumented and vulnerable to exploitation [15].

There are three ways for children to acquire a nationality or citizenship: (1) right of blood (jus sanguinis), where nationality is obtained through one or both parents; (2) birthright citizenship (jus soli), where nationality is obtained based on the country of birth; and (3) naturalization, where nationality is obtained through a process of legal immigration or marriage [16]. Accordingly, there are three categories of children who are stateless or undocumented: First, children are born in or brought into the country by parents with an irregular status or refugee mothers may have difficulty acquiring either parents’ nationality or can only obtain an irregular status [16,17]. Second, the lack of birth registration may be due to parents’ fear of their illegal status’s being discovered [16]. Third, children become undocumented immigrants after their legal documents become invalid while remaining in the country [17].

This categorization defines a person or a child’s stateless status on the basis of the law of each nation and refers to de jure stateless persons. In contrast, there are de facto stateless persons whose nationality may not be deprived but who do not have access to preserve the right and protection of their nationality [18]. Both de jure and de facto stateless persons experience similar challenges, including uncertain legal identity, family separation, unstable living conditions, forced movement, exploitation, discrimination, social exclusion, and limited resources to basic welfare [16,18].

Unlike stateless children in other countries as refugee children or new immigrant children, undocumented children in Taiwan have unique challenges in both acquiring a legal nationality and qualifying for social welfare. Most of them fall into the de facto group because Taiwan operates the jus sanguinis nationality policy, and children of MWs should claim the “nationality of their mother.” However, many children become stateless because their mothers are UMWs who have limited contacts with formal support and thus decide not to register the birth as a nationality of the original country, or they even abandon the child. These undocumented children are often called “black babies” or “invisible children,” owing to their undocumented status and/or their Southeast Asian origin. They have darker skin; they lack an identity, a nationality, and right- or welfare-related qualification; and they are unknown to others, formal service sectors, and the government and thus appear invisible to society. Similar to de jure and de facto stateless persons, they have limited access to appropriate healthcare, childcare, and education. The Taiwanese government has recognized their needs and begun to respond to this issue. The following discusses related services around the world.

### 1.3. Services for Migrant Mothers and Undocumented Children

The rights of MWs and their families have been internationally recognized. The United Nations passed the International Convention on the Protection of the Rights of All Migrant Workers and Members of Their Families (CRMW), which requires all state parties to protect the rights of MWs and family members [14], including education, vocational training, social and health services, and cultural activity participation. Taiwan, however, is not a signatory of the CRMW and has not developed a clear policy to address the concerns and preserve the rights of migrant workers and their families.

The UNCRC also indicates that state parties should provide appropriate assistance and protection in response to preserving and recognizing children’s identity, nationality, and family relations [19]. However, the Taiwanese social service sector has been challenged because, owing to their undocumented or stateless status, children of UMWs do not qualify for Taiwanese social welfare. Thus, undocumented children’s unique needs require specific assistance.

Though services for undocumented children are underdeveloped in Taiwan, other countries practice different levels of support to ensure the best interests of children and migrant mothers [13,17,20,21]. However, undocumented children are often excluded from social rights in receiving primary healthcare and education. For example, countries such as Austria, Germany, and the United Kingdom provide limited health services to undocumented migrants and their children, usually for emergency care or limited medical conditions [17,21]. Other countries, such as France, the Netherlands, and the United States, offer special care systems for undocumented migrants and children [13,21]. Many illegal migrant parents may be fearful of being identified when seeking healthcare. For example, one study explored the health needs of undocumented children in Denmark and found that healthcare services for this population were underused. Reasons for the concern included fear of being criticized, lack of knowledge about healthcare availability, and lack of access to specific healthcare setting. In addition, the United States has limited welfare qualifications for immigrants: only those with “qualified” immigrant status for five years are eligible for most social welfare, such as Medicaid, Social Security, State Children’s Health Insurance Program, etc. [22]. Thus, a lack of healthcare, such as regular checkups during pregnancy and immunizations after birth, can negatively impact children’s early development.

In terms of primary education, many countries (e.g., Canada, France, Germany, Japan, the United Kingdom, and the United States) preserve the right of children to education regardless of their parents’ immigration status [20]. Undocumented children in these countries have access to free and compulsory primary education. However, this does not include free school meals, transportation, financial assistance for uniforms, or other expenses, which are critical supports for migrant children [17]. For example, undocumented children in the US experience issues related to schooling, such as poverty, language barriers, cultural differences, and multiple moves. These challenges create more barriers to the basic right of education for undocumented children [23].

In Taiwan, the issue of undocumented children has been apparent since the 2000s, but it was not until 2017 that the role of the Taiwanese government became clearer and critical to protecting their rights by developing practical principles and regulations. In 2017, the MINIA issued the “Guideline for Issuance of Alien Residence Certificate for Foreign Dependent Children,” which provides local governments with a practical procedure and ensures children’s right to a legal identity. According to this regulation, newborns can obtain legal documents based on their mothers’ nationality, and those whose parents are missing can be naturalized and later adopted domestically in Taiwan or to other foreign countries.

In 2018, the Ministry of Health and Welfare issued another regulation: once the child has obtained an alien residence certificate, they are qualified for coverage under the national health insurance program. Before this regulation was enacted, foreign children were covered by national healthcare only when they or their parents had legal immigration status. The development of this regulation is another milestone to preserving children’s right to appropriate healthcare, including routine vaccinations, physical checkups, and development assessments.

However, the protection of undocumented children’s developmental rights is still unclear, including appropriate primary education and childcare. Because students’ enrollment status is based on household registration and because undocumented children are not legally registered in a household, in practice, primary education is not guaranteed for all undocumented children. In addition, many undocumented children are considered unaccompanied and are involved in the child welfare system because their mothers are often UMWs. Although these children may receive appropriate childcare on a residential basis, they may not receive adequate service for permanent residency.

In sum, the rights of MWs and their children, either legal or illegal, are not fully protected in Taiwan. Because of their uncertain legal identity, they are likely to be deprived of various rights (e.g., rights to survival, protection, development, nondiscrimination, etc.). Although the literature on other countries has provided examples of migrant worker services, issues regarding UMWs and undocumented children in Taiwan are complex and unique and need collaboration between multiple systems and policies, such as the provisions of foreign affairs, education, labor, health, and welfare. However, past research on undocumented children in Taiwan has mostly been conducted from a legal perspective rather than from a human service perspective that focused on their rights to appropriate supports and services.

Thus, this study explores the precarious situation and service access of UMWs and their undocumented children, through interviews with stakeholders from multiple systems. The findings of the study clarify the needs and challenges of this population and raise concerns for better social service development. This study has the following specific aims:(a)To identify professionals’ views the needs of UMWs and their undocumented children.(b)To explore the healthcare and childcare service access for UMWs and their undocumented children from the views of professionals in related systems.(c)To understand current service access limitations among UMWs and their undocumented children, for practical implications.

## 2. Methods

This study is based on a qualitative research design, implemented between August 2019 and May 2021. To preliminarily explore the precarious status of UMWs and their children as well as identify the accessibility and availability to social services, this study involves perspectives of professionals from multiple systems. A purposive sampling approach was applied because in Taiwan there have been no consistent practices or concrete policies across counties/cities addressing the issue of UMWs and undocumented children. Because many UMWs were located in one city in which most of their undocumented children were placed, the current study is designed to explore this specific city’s practical experiences and service development in response to service accessibility and availability. In addition, we have attempted to reach professionals who are the most experienced in working with UMWs/undocumented children and relevant issues; thus, the recruiting process led to limited samples of people who are qualified to discuss the current inquiry. Accordingly, in-depth interviews were conducted with 12 stakeholders, including social workers at a specific residential care placement and a large hospital; administrators at the local government, a migrant worker detention center, and a foreign country’s economic and trade office; and staff at a mosque (see Table 1 for participant descriptions).

Data collection with multiple professionals can provide a preliminary understanding of UMWs and undocumented children’s challenges and needs. It is important for service providers to raise concerns regarding the limitations on healthcare and social service systems. Semistructured interviews were based on an interview protocol that includes basic questions for all types of participants and specific questions corresponding to different contexts. We collected perspectives from all informants, such as (1) experiences working with UMWs (e.g., MWs’ reasons for leaving legal employment, lives and resources after their leaving, challenges in meeting personal and children’s needs), (2) challenges experienced under unclear regulations, (3) strategies developed to protect UMWs/children’s rights, and (4) policy and practice suggestions for responding to the trend. Specific interview questions addressing concerns happened in different contexts, such as medical needs, caregiving needs, permanency goals, and religious and cultural factors that could be impactful on practical decisions. Interviews were conducted face to face and audio-recorded, though two interviews were conducted via online meeting owing to the COVID-19 pandemic. Data were transcribed verbatim and evaluated for themes [24]. Interviews ranged in length from 1 h and 50 min to 2 h and 30 min. Our data analysis involved a series of analytical discussions. First, open coding techniques are adopted to analyze interview transcripts. Second, conceptual labels on responses that described unique or repeated events, experiences, difficulties, and challenges reported in the interviews are positioned. Third, each individual interview across all questions is examined to identify metathemes. Finally, responses to common metathemes across all interviews are scrutinized. In addition, the research team constantly discusses themes that emerge from a thematic analysis.

## 3. Results

Findings from the analysis revealed current challenges to UMWs and their undocumented children, including (1) medical issues (e.g., nonattendance for the periodic prenatal exam and newborns’ special health treatments) and healthcare accessibility (e.g., lack of newborn vaccinations), (2) child placement (i.e., residential care in the foster care system), and (3) the permanency of undocumented children. The following sections demonstrate the major themes of the current findings.

### 3.1. Health Issue and Access to Healthcare

Female MWs can become pregnant before or while missing, and according to one social worker at a child welfare placement (residential care) center, the proportion is about equal. In either situation, choosing to be disconnected and take on an illegal status exposes them to more risks, particularly because pregnancy and giving birth require healthcare services. The following demonstrate challenges related to healthcare, and Figure 1 demonstrates different situations under which UMWs become pregnant and the challenges they encounter.

Pregnant MWs may have few options in terms of healthcare settings. To avoid being discovered, many UMWs tend to give birth at small clinics referred to them by other MWs. Others who may be in poor health during pregnancy have no choice but go to large hospitals to receive better medical care. To avoid being reported, some may leave the hospital without notice during treatment or after childbirth. However, it is detrimental to their recovery. The leaving behavior also excludes the newborns from undergoing medical checks and can lead to further health problems for both mothers and babies.

*In Islam, birth control is not allowed or encouraged. After becoming pregnant, MWs worry that their employer will discover it, so they will hide it and give birth secretly. That also increases the risk of stillbirth. Married female MWs may be afraid of returning to their home country because they have a relationship in Taiwan that violates their religious doctrine*.(Mosque staff, interviewed on 16 April 2020)

*UMWs are worried that if they go to a large hospital, they will be easily caught, and then not allowed to work in Taiwan and deported. So, they are likely to go to small clinics and even do not have prenatal exams. Therefore, some children’s diseases could not be identified earlier. Women would still have health risks due to sudden hormonal changes after giving birth. Although migrant mothers have been informed about their health condition, they may still decide to leave soon after childbirth*.(Hospital social workers, interviewed on 3 May 2020)

The above quotes indicate that UMWs tend to avoid large hospitals, although this is where both mothers and children could receive better medical care. Moreover, even when children are born at hospitals, they may become undocumented. Without legal status, there can be delays in urgent medical treatments, causing challenges for hospital social workers:

*Large hospitals are often the last resort for children with serious illness. Many children are sent to us because only we can provide special treatments. When a child has serious illnesses that needs to be addressed urgently, preparing the necessary documentation often delays the medical treatment*.(Hospital social workers, interviewed on 3 May 2020)

In addition to choices of the childbirth setting, healthcare costs limit pregnant UMWs’ choices of healthcare. UMWs are often not covered by health insurance and thus tend to give birth privately (such as at home or in a dormitory) because of the high cost of childbirth (around NTD $20,000 to $30,000, which is about USD $700 to $1080). Currently, most UMWs’ childbirth expense is supported by external charitable donations to the hospital, though there are also special cases. For example, when the newborn is left at the hospital by a missing migrant mother, the hospital will file a child protection report, and healthcare expenses will be reimbursed by a local government. However, if any local government cannot provide the reimbursement, hospitals will ultimately bear the expense, which can be challenging. Thus, healthcare issues among UMWs and undocumented children challenge not only this vulnerable population but also the public medical system.

For undocumented children, health insurance limitations excluded their access to developmental healthcare. Unlike other Taiwanese newborns, undocumented children did not receive their newborn vaccinations (e.g., pneumococcal vaccination is obligatory for babies under 1 year) and routine health checks before 2018 (as mentioned above, undocumented children can be enrolled in the national health insurance program under regulations issued in 2018). Hereafter, some undocumented children’s deaths occurred in 2017 and 2018 at child placement facilities owing to the lack of newborn vaccinations. The incident also discovered childcare concerns at the residential care placement, which will be discussed.

### 3.2. Childcare and Child Development

Many undocumented children are left at a nonprofit organization that provides childcare as a residential placement because their mothers need to work after childbirth and cannot care for their child. This placement service is well known to migrant mothers and is the only organization caring for undocumented children. However, this institution was not licensed or supervised by the local government until 2019, leading to challenges for placement workers.

First, because the placement was previously not regulated by the government, case workers were often overloaded. With limited space, the quality of childcare was often reduced. However, even though the challenge was known, the local government had to accept the ambiguous status of this facility because this facility was the only organization caring for this particular population.

Moreover, placement workers might struggle with acting in the conflicting roles of social support (i.e., childcare provider) or social control (i.e., mandatory reporter about the issue of unaccounted status). From the perspective of local government administrators, it would be beneficial if UMWs and their undocumented children could receive formal help whenever possible, rather than using informal resources. This would also ensure the rights and best interests of undocumented children.

Despite these challenges, the city government where the original nonprofit organization offers services for undocumented children has developed a standard procedure and child placement practice. This public–private partnership promotes the registration of residential children, so childcare for undocumented children is currently supervised and ensured. In addition, the local government can file the child cases and work to return both mothers and children.

Furthermore, the theme of trust in the relationships between the government, the placement facility, and migrant mothers is very important. Workers at the placement facility appear to be more trustworthy according to mothers because they not only help with childcare but also often share children’s photos and videos. Thus, mothers can be reassured by constantly monitoring their children’s conditions. On the contrary, the main goal of government workers for those UMWs and undocumented children is to return them to their country of origin within 1 year. However, many UMWs would choose to overstay because they need to earn more money to assist with finances at home. This decision could harm their children’s health, development, and adaptation because of separation from their mothers. Thus, the effective provision of childcare relies on gaining the trust of UMWs.

*At present, we serve as a bridge to connect mothers and local governments. … Some administrators in other counties may think that these migrant mothers would run away, take advantage of Taiwan’s social welfare resources, like to play, and are irresponsible. Indeed, many migrant mothers came to us in tears, saying that they are not allowed to visit their children, have difficulties applying for documents, or fear that they will be arrested if they do not return to their country soon. They [public administrators] are very unfriendly*.(Placement social worker, interviewed on 13 August 2020)

#### Child Development and Education

Study participants indicated that some undocumented children are found to have a developmental delay at the child placement facility, which can be attributed to a lack of mother–infant attachment and appropriate stimulation. Those UMWs are afraid of being reported and have no choice but to live and work secretly, sometimes looking after their children from a distance, limiting the development of appropriate mother–child relations. Those whose pregnancy and childbirth were not planned may even not inquire into their children’s situation until being caught and deported by immigration officers. A secure and trusting attachment may not be developed, and the mother–child relationship and interaction may be strained. The child’s general development, then, may be negatively influenced by the low quality of the mother–infant attachment and lack of individualized stimulation in a group-care setting.

*In some cases, the local government has been the official guardian of a child. But when the biological mother was found, and they must be deported together. Many children are very resistant and present symptoms of anxiety and bedwetting. The placement workers try to help these children by familiarizing them with their future living environment by providing photos of their mother and local communities in Indonesia, as well as teaching the language*.(Local government administrator, interviewed on 11 May 2020)

In terms of children’s education, unlike other children in Taiwan with legal residence status who thus qualify for primary education, undocumented children who are not identified do not have a nationality and cannot receive admission notices when they reach school age at 6 years of age. These children can enroll in public school as temporary students to receive formal education in Taiwan. However, they cannot obtain official student status, academic qualifications, or graduation certificates, which significantly reduces their rights to human capital development. Taiwanese primary schools also cannot provide culturally appropriate and individualized education to these children of migrants, so their right to cultural connection is also a concern.

### 3.3. Permanency Planning for Undocumented Children

“Permanency” can be defined as a stable and enduring status for children in out-of-home care [23]. Undocumented children in Taiwan are not considered foster children; they are temporarily placed in residential facilities and are expected to be reunified with their family and eventually return to their country. Thus, permanency planning is still important for these children but has not been properly developed. According to study participants, stateless or undocumented children can be generally categorized into two groups, according to their nationality. Figure 2 demonstrates the journeys of these children.

#### 3.3.1. Naturalization and Adoption

Due to Taiwan’s jus sanguinis nationality policy, some children’s nationality cannot be identified, so they are considered truly stateless status because their birth parents are unknown or missing. In order to preserve children’s right to legal identity and help them acquire a nationality, the following response describes the naturalization process:

*If the birth mother is missing, the National Immigration Agency will search within the Taiwan borders for six months. If the mother has left Taiwan, the Ministry of Foreign Affairs will search for three months. If children’s mother cannot be found, the Ministry of Interior will determine the child’s Taiwan nationality through the adoption and naturalization process*.(Local government administrator, interviewed on 11 May 2020)

After confirming that the birth mother has been determined to be missing and not traceable, local governments will place undocumented children up for adoption, either domestically or internationally. After an adoption has been completed, the child can acquire Taiwanese nationality (through the naturalization process) or another foreign country’s nationality.

However, permanency through naturalization and adoption can be challenging. Children may not be considered stateless if birth mothers have provided enough information to recognize the child’s nationality, even when the mother is still missing. In other cases, the mother is not missing but would like to relinquish the child for adoption. However, because their child has been identified as a nationality of another country, the Taiwanese government cannot address the adoption process for a foreign child. This presents a dilemma for either maintaining the mother’s nationality of origin or finding a permanent home for the child. Thus, one public sector staff member mentioned that “*it is easier [for us] to manage those whose mothers cannot be found.*”

#### 3.3.2. Repatriation (Voluntary Departure) with Birth Parent(s)

Unlike children who are recognized as stateless, some children can acquire their mother’s nationality. These children are often expected to soon return to their countries of origin. Both local government and residential care social workers play important roles in encouraging regular mother–child meetings and proceed with the departure process. However, many migrant mothers, either legal or illegal, prefer to stay in Taiwan to make more money and expect the social welfare sector to continue providing childcare. Social workers then must emphasize that the residential care is temporary, and it is for “the best interests of the child” to provide childcare within their family and culture of origin.

In addition to the role of social workers, other administrators, such as the MINIA and the economic and trade office or economic and cultural office of foreign countries, cooperate to address issues regarding mothers’ illegal residence and children’s documentation. For example, a study participant from the local government indicated the following:

*Although children are under the guardianship of our local government, they do not have a Taiwanese identity. When they are prepared for departure, the National Immigration Agency helps prepare travel documents, and the foreign country’s economic and trade office helps verify these documents. Thus, after the mothers bring their children back to their country, they can claim the child’s nationality more easily*.(Local government administrator, interviewed on 11 May 2020)

Currently, because mothers of undocumented children are mostly migrant workers from a country of Southeast Asia, the country’s economic and trade office plays a more active and collaborative role. The office will contact the mother’s original family to ensure that they know about the mother and child’s return. The office also encourages those who are illegal residents but willing to return home with their children to self-report their appearance for penalty relief and air ticket reimbursement (i.e., through the overstayers voluntary departure program, executed by the MINIA).

However, practical issues may complicate the return process. Some mothers already have a family with a legal spouse and children in their countries, with the undocumented children being born out of wedlock in Taiwan. These mothers may not be willing to disclose the condition, and their birth family may not take responsibility for childcare. They may secretly leave their children to a friend or orphanage for adoption after going back to their country, which can be an unintended consequence of child permanency.

*The female MWs already have the child [in Taiwan]; if the spouse [in home country] cannot agree upon the reality, they would divorce. … Mostly they choose to divorce and ask their birth family to care for the child instead of sending the child to an orphanage. … Because the mothers would keep sending money back to the family … if they send the child to an orphanage, that is definitely for adoption*.(Foreign country’s Economic and Trade Office, online interviewed on 24 April 2020)

Another concern is that many mothers would prefer to stay in Taiwan to earn more money, although they know that without a legal identity, staying in Taiwan would delay their children’s development. Social workers are concerned that the longer the children remain in Taiwan, the more difficulties they may encounter. For example, many children have not yet developed a secure attachment with their mother, because the mother left the child in the placement since childbirth and has not been the primary caregiver. Although there may be arranged meetings, the child–mother relationship may not have developed sufficiently for their life after being deported. Thus, regarding the dilemma between the best interests of the child and those of the mother, there is always a timing question where the child’s development should be situated and how the mother can be involved in the planning process.

#### 3.3.3. Repatriation: Child Only

In 2018, the local government cooperated with one foreign country’s economic and trade office to develop a special program to prepare undocumented children for repatriation. This program solves the concern that many undocumented children stay in Taiwan longer than expected when their mothers extend their stay for more earnings. This solution also recognizes the principle that children will best develop in their family and culture of origin.

Often, the Taiwanese local government issues the child’s birth certificate, later verified by the Ministry of Foreign Affairs. Using the birth certificate and related information, the foreign country’s economic and trade office will contact the mother’s family. If a family member would like to care for the child, the office will arrange the child’s departure, and someone will take the child home after arrival. If no family member is willing to provide care, the office and local social welfare sectors will place the child in an orphanage after the child is sent back to their home country. This program, however, has been hindered by the COVID-19 pandemic in 2020 and 2021.

#### 3.3.4. Uncertain Journey and Permanency

Some government and placement social workers are concerned about whether sending undocumented children back to their countries truly meets children’s best interests given that these children may have an uncertain future after being repatriated. As indicated, many children have some preparation to reconnect with their mother and their culture of origin. However, they have been raised as most Taiwanese children would be: with localized approaches of care and education. Returning to their countries of origin could create challenges both in changes in the caring environment and in adaptation to a very different culture, which is not the culture that these children have been immersed in since birth. Thus, some study participants are worried about such uncertainty.

*These children would be very anxious. [wondering if] this person is my mom, and how do I get along with her? And I am going to leave Taiwan and go back to my original country … This is a big challenge for these children because their future is unknown. They don’t know about the community, the language, the culture … but according to the current Taiwanese law, they must go back to their country with their mother*.(Local government administrator, interviewed on 11 May 2020)

Additionally, unlike children who are adopted or discharged from the child welfare system in Taiwan, so that they may be followed afterward, the life of children who are repatriated to their country is unknown. The Taiwanese government has not developed any follow-up system to ensure that sending these children back to their country meets their best interests. Whether these children are raised and develop in a stable situation by their birth family cannot be forcibly tracked. A placement worker shared the following story:

*I think for now we are trying to do our best to care for the children, so that they [MW mothers] can securely work, save money, pay debts, and return home with their children. Fewer mothers are abandoning their children. We have heard that some mothers would abandon their children, or after going back to Indonesia, they would give or sell their children to other people or local orphanages in Indonesia*.(Foreign country’s economic and trade office, online interviewed on 24 April 2020)

Thus, it is challenging for service providers in Taiwan to assess children’s future best interests. There can be a dilemma between ensuring the child receives appropriate care in Taiwan and keeping the child connected with their original culture by sending the child back to their original country without any information on the resulting permanency.

## 4. Discussion

The current study reveals the precarious status of both UMWs and their undocumented children. In addition, service barriers are identified from multiple stakeholder perspectives. Our findings indicate that the Taiwanese government and nonprofit organizations strive to preserve children’s basic rights (e.g., rights to healthcare, childcare, education, cultural connection, etc.), but the social service and healthcare systems are still in need of further development.

### 4.1. Advocacy for Protection for UMWs and Undocumented Children’s Basic Rights

In Taiwan’s growing number of MWs, about half are women (53%) and at the age range of 25–34 years (48%) [1]. In their young adulthood, it is occasional to have unprotected sexual relations without contraception. However, pregnancies conflict with the interests of employers. Consequently, the numbers of UMWs and their undocumented children are increasing, leading to high risks (i.e., abortion, stillbirth, lack of basic care), and these people’s predicament continues [24,25]. The Taiwan government is moving toward developing protection for UMWs’ human rights and their children’s interests. However, the complexity of this issue comes from the intersections of multiple systems, including policies for migrant workers, immigration, health, and social welfare.

In terms of MW policy, MWs are often treated as temporary, low-skilled workers who cannot become citizens in the host countries, because of labor, political, social, and immigration regulations. As long as host countries do not realize that being missing is a way to respond to the deprivation of basic rights, these UMWs will continue to be treated as illegal and even criminal [25,26,27]. A similar condition has also been a concern in a nearby country, Japan. Migrant workers’ runaway acts often happened to “industrial trainees,” and they became unauthorized workers. In addition to seeking a better job with a higher salary and employment status (a common issue in Taiwan), studies indicated another major reason for the escaping decision is that the illegal network of migrant workers helps migrant workers overstay in Japan [28,29]. Thus, the contradiction between legal and illegal supports and between governmental regulations and rights protection for UMWs and their undocumented children is a significant social issue.

Although MWs are protected by the Act of Gender Equality in Empowerment, as are other Taiwanese laborers, their pregnancy rights are unattainable in practice [5]. Pregnant MWs are forced to choose between keeping their job and keeping the baby, leading to their missing behavior and their subsequent illegal, underground, and vulnerable conditions. Lack of healthcare, social welfare, primary education, and child placement in the community further harms UMWs and their children.

In terms of children’s right to development and the associated childcare system, the separation of undocumented children from their mothers can be harmful to their attachment, psychological development, and health. Children should have the right to stay with their parents, though it is challenging for UMWs to be responsible for work and childcare when they lack support. In particular, the current findings uncovered the fact that most undocumented are actually not stateless children but just “non-Taiwanese” children temporarily without legal status in Taiwan. In addition, their mothers are not guestworkers and eventually will go back to their original countries. This makes the issue in Taiwan different from that of European countries, such as the UK, in which reasons of statelessness often derive from children’s being born to refugee immigrant parents [22], or Asian countries, such as Malaysia, in which migrants pursue citizenship [30,31]. Because these children are able to acquire a legal nationality of another country, Taiwanese workers, caregivers, and administrators’ only goal is to deport the child to their home country as soon as possible. However, many mothers are sometimes disconnected and leave these children unaccompanied, creating more challenges for professionals to determine whether the case planning of the child or welfare policy should be made for a short-term or a long-term basis.

Although current findings indicate that access to healthcare and social welfare services is developing, structural obstacles and systematic discrimination still exist. For example, local hospitals and governments are able to cover the prenatal care cost to prevent stillbirth and any health problems. However, these mothers are still afraid of being caught and deported when they are missing. This is like a continual feeling of imprisonment in precariousness, where they cannot have a normal life in this marginalized condition [17]. In short, the reasons for being unaccounted for and the social beliefs about MWs in Taiwan are related to how Taiwanese policies, systems, and people recognize the basic rights of MWs and their children as human beings. Thus, further policy analyses in Taiwan and other countries can help to examine and protect the basic rights of pregnant MWs and undocumented children.

### 4.2. Advocacy for Culturally Appropriate Practice

The current findings reveal some practical strategies developed by local public and private sectors. Unlike the interviewed social workers, who have established partnerships between public and private sectors and built trust relationships with UMWs, other practitioners without experience in working with female UMWs and undocumented children may not acknowledge this vulnerable population’s needs, challenges, migration history, life while being missing, and reasons for overstaying despite being illegal. These staff members may also not be familiar with the social and cultural context of MWs’ running-away behaviors in Taiwan. Moreover, the primary goal of the public sector is often to send undocumented mothers and children back to their country. If practitioners are not aware of cultural factors (such as religious and family values) associated with the decision-making in UMWs’ overstays or returning, they might consider MWs’ missing behaviors as an illegal/criminal action and further give an authority cause to deprive MWs of their rights. Thus, training on MWs’ cultures, values, and life styles, along with developing culturally appropriate practices with female UMWs and children, is critical for staff in Taiwan’s human service sectors. In this way, UMWs’ struggles between making a living and providing childcare, as well as the dilemma between overstaying in Taiwan and returning to their home country, can be acknowledged and effectively addressed in a supportive environment.

In addition, the major reason for female MWs’ becoming missing and having undocumented children is unplanned pregnancy, and thus, preventing pregnancy could be an effective solution. However, pregnancy prevention or birth control is concerned a cultural issue. Some behaviors of MWs, such as premarital/extramarital sex and using contraception, conflict with their religious doctrines and ethical beliefs. However, their beliefs and values may change over time, especially in a foreign country with unfriendly attitudes and discriminations, which may create internal conflicts in MWs that are not recognized by Taiwanese people, leading to more misunderstandings.

To prevent unexpected pregnancy, MWs, especially men, are encouraged to have safe sexual behavior by following appropriate birth control methods. According to study participants, currently, some religious centers strongly advocate for safer sexual behaviors by disseminating booklets during religious festivals at places where MWs are likely to gather. Furthermore, parental education is being developed for all MWs.

### 4.3. Advocacy for Interdisciplinary, Interministerial, and Transnational Collaboration

According to current findings, issues regarding UMWs and undocumented children are complex, and there is no one specific system that can comprehensively address these concerns. With the growing number of UMWs and undocumented children, the most important task is to understand the multilevel and multisystem factors causing migrant workers to become missing. The current findings provide a range of views from different sectors. Most of them argued that it is important to involve efforts from all sectors related to MWs (e.g., migration, labor, healthcare, childcare, education, etc.), and all issues involving these sectors are interconnected. The collaboration should also be interministerial and transnational because the social issue involves migrants and children with multilevel factors and from several nationalities. Thus, further exploration into the collaboration among different systems is necessary, which will raise public concerns for protecting the rights and interests of MWs and their families through social welfare systems.

### 4.4. Limitations and Implications for Future Research

The current study has some limitations. First, although the data collection with various stakeholders from different systems provides nuanced perspectives to explore and deliberate on the vulnerable situation of UMWs and their undocumented children, most interviewees were administrators and professionals at sectors in one specific city in Taiwan. Although these service providers are experienced in addressing issues regarding UMWs and undocumented children in Taiwan, applying this study’s findings to another context (i.e., transferability) must be performed cautiously. Second, this study involved small sample of 12 participants and cannot be generalized to the population of focus, as a limitation. As mentioned, these professionals/stakeholders were recruited because they were well experienced in and knowledgeable of the issues of UMWs and undocumented children. Despite the limited sample, findings from this study will hopefully raise concerns of the social issue and welcome more professionals and scholars to advocate for vulnerable populations. Future studies should consider increasing the sample size as well as recruiting different voices. In particular, subjective views of the UMWs themselves are absent in this research. Especially during the COVID-19 pandemic, UMWs with children concealed in the community may have higher risks. Future research can include voices of MWs or UMWs to fill this gap in the literature. Third, this study considers recent policy changes that protect the basic rights and well-being of this population. There are a few older children who went through different experiences of being undocumented or stateless a decade ago and for whom identity and education are issues. Their situation is unknown and needs further exploration. Lastly, children’s well-being after being deported is not tracked. Whether they achieve permanency and live in a safe environment with nurturing relationships is a concern. To ensure the best interests of the children, future research can further evaluate the effectiveness of current approaches and feedback can inform future policy reforms.

## 5. Conclusions

Issues regarding UMWs and undocumented children are intertwined, and there is no one specific system that can comprehensively address these concerns. With the growing number of UMWs and undocumented children, the most important task is to understand the multilevel and multisystem factors causing migrant workers to become missing. The precarious status of UMWs and their undocumented children is derived from the absence of legal citizenship status, nationality documentation, and national health insurance coverage. This article explored the vulnerable situation of UMWs and their children, who are suffering in precarious conditions, and it explored their challenges through interviews with multiple professionals. The preliminary findings indicated a fundamental issue: UMWs and children’s rights were affected by multilevel factors (family, employer, agent, nation, religion, and transnational factors) and multiple policies (migrant worker, labor, social welfare, healthcare, and immigration policies). Although the right to pregnancy for female MWs is protected by law, it is not permitted in practice and leads to workers’ becoming missing. Further issues regarding healthcare, childcare, and child permanency are addressed with limited resources and deserve much more attention.

To conclude, becoming pregnant for female MWs should not be considered a problem but rather a normal choice. The issue not only is detrimental to both mothers and children but also imposes social and economic costs on the Taiwanese government. Therefore, intense collaborations between private and public sectors, interministerial efforts, and transnational governments that adopt a multilevel approach will be the next important strategy to respond to this issue and improve the well-being of UMWs and undocumented children.

## Figures and Tables

**Figure 1 ijerph-20-00956-f001:**
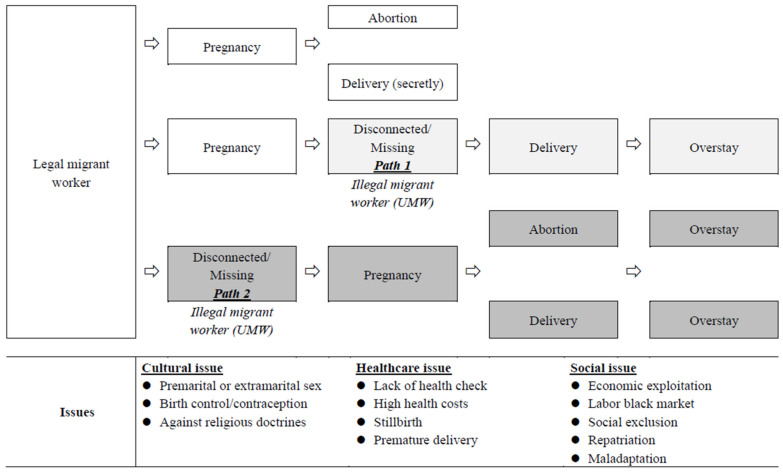
Paths of legal female migrant workers being pregnant and becoming unaccounted for migrant workers (UMWs) Various stages.

**Figure 2 ijerph-20-00956-f002:**
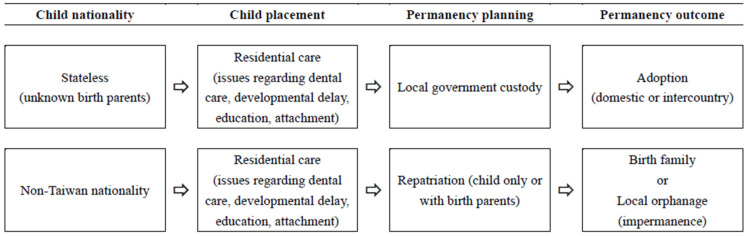
Permanency/placement planning for children of migrant workers in Taiwan.

**Table 1 ijerph-20-00956-t001:** Interviewee description.

Professional Position	Role	Number of Interviewees
Local government (social welfare sector)	Administrator	3
Child welfare placement (residential care)	Social worker	3
Migrant worker detention center	Administrator	2
Foreign country’s economic and trade office	Administrator	1
Hospital	Social worker	1
Mosque (religious center)	Staff	2

## Data Availability

The information from this study is in the hands of the researchers and can be made available to the journal, respecting the anonymity of the respondents, if its editors deem it necessary.

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
