# Peer review of "Barriers to Health and Social Services for Unaccounted-For Female Migrant Workers and Their Undocumented Children with Precarious Status in Taiwan: An Exploratory Study of Stakeholder Perspectives"

_ijerph, 2023, doi:10.3390/ijerph20020956_

Round 1

Reviewer 1 Report (Previous Reviewer 2)

The sample size is far too small and the revisions, such as they are, do not address this. Calling it an exploratory study doesn't really change anything.

Author Response

Point 1: The sample size is far too small and the revisions, such as they are, do not address this. Calling it an exploratory study doesn't really change anything.

Response 1: We really appreciate for the reviewer’s comment raising the concern about the small smaple size in the study. We have acknowledged this issue as one of the study limitations. But we do want to raise the public concern and futher government response of the social issue by emphysizing its significance. Because the current acess to the health and social services for those unaccounted for female migrant workers (UMWs) and their undocumented childern are underdeveloped in Taiwan, the purpose of this article is to explore the barriers to health and social services for this vulnerable population. During the past two decades, there was only one social welfare non-gonvermental orgnaization well-known to UMWs. Because this NGO has been experienced working with issues related to female migrant workers and undocuemented children and it is located in a northern city in Taiwan where most UMWs would gather together and share the information when needed. Particularly, this NGO is well-knowen to female migrant workers for its residential care for their newborns, and for now, this child residential care is the only liscened child welfare placement caring for the most non-Taiwanese children in Taiwan. Thus, governmental and private setting workers and administrators in the northern city make a small group, but they do represent the most experiened stakeholders working on this issue and providing related services and care for those UMWs and undocumented children. Thus, few stakeholders were recruited in our research, resulting in the small sample size, as indicated in line 215-230.

Although with the small sample size, this study, as we hoped, will to raise the concerns of the current issue and trend and to welcome more professionals to join us advocating for the vulnerable population. We do understand that subjective views of the UMWs themselves are absent in this research. We had already mentioned as our limitations in the part of 4.4. Limitations and implications for future research (in line 652-662). We have proposed to conduct another following research on the dynamic process and subjective interpretation of migrant workers’ current living and precarious, and hopefully the futher research will involve as much more samples and voices.

We definitely understand that some scholars consider the more samples the better or stronger evidence. But there are also other scholars (Patton, 2002; Marshall et al., 2013; Morse et al., 2002) emphasizing on “purposeful strategies” instead of “methodological rules” or on “the depth of data” instead of “the number of sample size.” That is, the depth of data may be better than frequencies to measure sufficiency of sample size. Because qualitative inquiry with a high tolerance for ambiguity, the sample size rule will be contingent on “what you want to know, the purpose of the inquiry, what’s at stake, what will be useful, what will have credibility, and what can be done with available time and resources” (Patton, 2002, pp. 242-243). It is more important to sample the best participants who best represent the research topic (Morse et al., 2002). As mentioned above, the purposive sampling conducted in our study has helped us reaching the most experienced and best representative workers regarding the social issue. We considered increasing the sample size within the context may not guarantee the increase in the richness or quality of the data. The reference lists as below.

Marshall, B., Cardon, P., Poddar, A., & Fontenot, R. Does sample size matter in qualitative research?: A review of qualitative interviews in IS research. Journal of computer information systems2013, 54(1), 11-22.

Morse, J., Barrett, M., Mayan, Mayan, M., Olson, J., & Spiers, J. Verification Strategies for Establishing Relibility and Validity in Qualitative Research. International Journal Qualitative Methods, 2002, 1(2), 13-22.

Patton, M.Q. Qualitative Research & Evaluation Methods. Sage, Thousand Oaks, CA, 2002.

Reviewer 2 Report (Previous Reviewer 1)

 I have found the following typos that the author's need to correct before publishing the article.  

Corrections are required:  

Line 9 - Drop the dash "Unaccounted for".   

Line 29 - It should "place of employment" and not "placement of employment."  

Line 30 - It is "ran" and not "run." 

Line 31 - It is "... referred to as ..."  

Line 35 - "vulnerability: ..." and not "vulnerability, "  

Line 50 - Drop the dash. ( See 9 above.)  

Line 52 - "to work legally"  

Line 58 - "reasons for ..."  

Line 63 - "referred to them"  

Line 73 - Drop "to be".  

Line 128 - Delete the spaces between the quotation mark and 'The.'  

Line 424 and 425 - "maintaining the nationality of the origin" does not sound right. Perhaps it should be "maintaining the mother's nationality of origin."?  

Line 432 - "proceed with the ..."  

Line 519 - It should be "their" and not "the".  

Line 665 - Add a space to "next important strategy."  

If these corrections are made then the article can be published.

Author Response

Point 1:  I have found the following typos that the author's need to correct before publishing the article.  
Corrections are required:  
Line 9 - Drop the dash "Unaccounted for".   
Line 29 - It should "place of employment" and not "placement of employment."  
Line 30 - It is "ran" and not "run." 
Line 31 - It is "... referred to as ..."  
Line 35 - "vulnerability: ..." and not "vulnerability, "  
Line 50 - Drop the dash. (See 9 above.)  
Line 52 - "to work legally"  
Line 58 - "reasons for ..."  
Line 63 - "referred to them"  
Line 73 - Drop "to be".  
Line 128 - Delete the spaces between the quotation mark and 'The.'  
Line 424 and 425 - "maintaining the nationality of the origin" does not sound right. Perhaps it should be "maintaining the mother's nationality of origin."?  
Line 432 - "proceed with the ..."  
Line 519 - It should be "their" and not "the".  
Line 665 - Add a space to "next important strategy."  
If these corrections are made then the article can be published.

Response 1: We really appreciate the reviewer’s detailed suggestions in a careful and patient manner as well as wonderful recommendations which improve the quality of our article. We have went throught the whole article and revised typos as the following.  
Line 30 – We replace "run" as "ran."
Line 31 – We add "as" become... "referred to as ..."  
Line 35 – We change multiple "vulnerabilities, " into multiple "vulnerabilities: ". 
Line 52 – We change as "to work legally"  
Line 58 – We replace "of " as "for" become "reasons for ..."  
Line 63 – We add "to" become "referred to them"  
Line 73 – We drop "to be".  
Line 128 – We delete the spaces between the quotation mark and 'The.'  
Line 424 and 425 – We revise as "maintaining the mother's nationality of origin."  
Line 432 – We add "with" become "proceed with the ..."  
Line 519 – We replace "the"interest as "their" interest.  
Line 665 – We make sure a space in the "next important strategy."  
The above recommended corrections are made.

For the following, we did not make changes and reasons are provided here:
Line 9 and 50 – We did not drop the dash “unaccounted-for” since “unaccounted-for migrant worker” is a term used in our governmental official document. We followed the exactlly use of the term to avoid confusion and hope the consistency will ensure related academic publications can be found.

Line 29 – We did not use the term “place of employment” because migrant workers are often “placed” or “assigned” to a work place. Migrant workers are not free to choose, leave, or change a jobs. Thus, we use the term “placement of employment” referring not just a physical work place they work but the employment they were arranged and placed.

Reviewer 3 Report (New Reviewer)

The authors of the manuscript dive into barriers to  health social services for illegal migrant workers in Taiwan, an important and highly relevant topic that is well described and reasoned throughout the manuscript. The sections are well written and the thread is linear and clear from introduction to conclusions. 

Some minor comments for consideration are:

1. The authors describe several stakeholders, relevant for the topic under study. It seems from the analysis  that many new-borns are left by their mothers because of employment reasons. Have the authors considered interviewing either employer organizations, or a labour market expert or anyone relevant in the area in Taiwan that could give an insight on what could be done from labour market policies in this regard?

2. Under the section 4.1 the authors state "In their young adulthood, it is common to have romantic relation-543 ships that include sexual relations.". I would delete such sentence since it seems to force a romantic connotation to sexual relations. 

Major comment:

1. While the authors describe very well the situation and the results in Taiwan, is lacking a bit of international context and comparison both in the introduction and the discussion of the results. What is already known on the topic form previous literature? How do the results compare to exisiting findings (if any) either internationally or to neighboring countries with a similar system or problem? 

Author Response

Point 1: The authors of the manuscript dive into barriers to health social services for illegal migrant workers in Taiwan, an important and highly relevant topic that is well described and reasoned throughout the manuscript. The sections are well written and the thread is linear and clear from introduction to conclusions. 

Some minor comments for consideration are:

(1)The authors describe several stakeholders, relevant for the topic under study. It seems from the analysis that many new-borns are left by their mothers because of employment reasons. Have the authors considered interviewing either employer organizations, or a labour market expert or anyone relevant in the area in Taiwan that could give an insight on what could be done from labour market policies in this regard?

(2) Under the section 4.1 the authors state "In their young adulthood, it is common to have romantic relation-ships that include sexual relations.". I would delete such sentence since it seems to force a romantic connotation to sexual relations. 

Response 1: We sincerely appreciate the reviewer’s encourgements and compliments as well as the recommendations which help us to provide more insights and supplement to illustrate the context. The following addresses the reviewer’s minor commets:

(1) As the reviewer clearly pointed out, the working conditons, the forbidden free transfer to another employers, agent fee, and the disqualified of labor law protection from the labor policy (for those live-in migrant care workers) are crucial factors which affect the the lives and right-related issues of unaccounted-for female migrant workers and their undocumented children (which is discussed in line 58-76). The major reason we did not consider interviewing employer organizations or labor market experts is that this article put more emphasis on the barriers to health and social services for UMWs and undocumented children rather than on migrant workers’ labor issues or worker-employer relations. Besides, the purpose of the female UMWs left their newborn babies included continuing earning more money in Taiwan to pay the debt or/and sending the remittance back to family in their home country. It would be challenging to interview the employers or employer organizations who hired those UMWs illegally. Also, we considered stakeholders realted to labor issue may not play an important or supportive role throughout the running away and pregancy process. However, we do consider continuing our research on how employment, immigration and care regimes/policies affect the precariousness of all migrant workers and lead to disconnected/runaway phenomenan. We really appreciate the reviewer’s recommendations.

(2) We have revised the statement "In their young adulthood, it is common to have romantic relationships that include sexual relations" as "In their young adulthood, it is occasionally to have unprotected sexual relations without contraception". (line 551-552)

Point 2: Major comment:

  1. While the authors describe very well the situation and the results in Taiwan, is lacking a bit of international context and comparison both in the introduction and the discussion of the results. What is already known on the topic form previous literature? How do the results compare to exisiting findings (if any) either internationally or to neighboring countries with a similar system or problem? 

Response 2: We have mentioned the UN conventions in the introduciton (line 133-154, 161-168) and included/added more contexts regarding other countries’ social services for migrant workers and undocumented children (line 155-161, 168-171). We have also reemphasized the uniqueness of the issue in Taiwan by comparing the staus with other countries in the discussion section [line 564-571, 583-594].

This manuscript is a resubmission of an earlier submission. The following is a list of the peer review reports and author responses from that submission.

Round 1

Reviewer 1 Report

An interesting research topic and study that looks at a very complex and difficult public policy concern for Taiwan and, I suspect, many other countries.

I have a number of concerns with this submission as it presently stands. These fall into broadly three categories. The first are methodological and the second is the analysis of the qualitative interview data. And, the third is the quotation and citation of interview data.

With respect to the methodology, the study is based on 12 interviews with administrators, social workers and staff that seem to be located in government departments and agencies and a religious centre. It is unclear how these people were selected for the interviews. Moreover, interviews based on a mere 12 respondents raises questions regarding how representative this sample is of all those who are working with female migrant workers, unaccounted or not, and their children. There are also concerns regarding the "semi-structured interviews." Given the range of organizations involved were any of the questions asked the same for all the respondents or did they vary? If some of the questions were standardized then what were these questions? Since the research was conducted with human participants was ethics approval sought and was it granted to proceed with this qualitative study of government social workers and administrators and staff at a Mosque?

What also appears to be missing from this study is the acknowledgement of the inherent biases of the sample of informants whether they are professional biases of State officials or the disciplinary biases of social workers or administrators. This could impact the study in various ways, particularly, given the small size of the sample involved.

Related to this is the question of the relative size of the population of concern, that is, the estimated number of unaccounted female migrant workers with children. I do not recall whether any statistics were presented in this regard. While the different patterns were outlined with respect to those who choose to abandon their children or to return to their countries of origin, etc., I do not recall whether any estimates were provided of the numbers in these various pathways of the mothers and their children. Admittedly, this might be extremely difficult to estimate properly but overall this might help to provide a sense of the most prevalent ways in which the mothers and their children respond to their circumstances. This would have an obvious bearing on how one might wish to respond to these concerns through the appropriate public policy meansures. And, related to this is how many unaccounted female migrant workers are there without children, that is, never had a pregnancy?

With respect to the anaylsis of the interview data gathered it was unclear how the quotes that were selected were derived. There was some mention of continually discussing themes that emerged from the interviews but it was unclear why these specific quotations were selected. The quotations from the interviews were in italics but are these direct quotations or are they some refined quotation based on a compilation of several interviews? Direct quotations should be in quotation marks and the sources need to be specified such as (social worker, interviewed on June 21, 2020). For some quotations there was an effort at identifying the source but not at others. How long were the interviews on average? How were the audio recordings and transcriptions of the interviews stored securely, along with the consent forms of the respondents?  

I found the analysis of the interview data quite limited. There is no indication of whether there were the same or similar responses from those interviewed on any standardized questions. Was there any noticeable variance among those interviewed? One might assume that administrators might have different views than social workers, as an example, given the issues under consideration. Likewise for the staff who were interviewed at the Mosque.

A further comment on the findings. My sense was that there was little engagement here with the literature on this topic and/or with government policies, whether local or national. 

The suggestion that further research with female migrant workers with children, unaccounted or not, is, I think, essential before offering any concrete policy recommendations on this subject.

And an obvious limitation of this research study is the small size of the sample. It would be hazardous to offer any definitive recommendations based on these interviews alone.

I also found the following errors in the text:

Line 92        Should it not be "substantially."

Line 127      It is UMWs and not UNWs. 

Line 165      There is an "r" missing in "counties" that should be "countries."

Line 309      Should it not be "concerns."

Line 342      A word seems to be missing. Is it not, "This decision "could" harm ..."

Lines 385 and 386   Should there not be a citation here when defining the term "permanency"?

Author Response

Response to Reviewer 1 Comments

Point 1: An interesting research topic and study that looks at a very complex and difficult public policy concern for Taiwan and, I suspect, many other countries. I have a number of concerns with this submission as it presently stands. These fall into broadly three categories. The first are methodological and the second is the analysis of the qualitative interview data. And, the third is the quotation and citation of interview data.

Response 1: We really appreciate the reviewer’s wonderful suggestions which carefully point out the inadequacies of our content and even typo in detail which will improve the quality of our research. We carefully track the reviewer’s comments to make change as the following points.  

Point 2: With respect to the methodology, the study is based on 12 interviews with administrators, social workers and staff that seem to be located in government departments and agencies and a religious centre. (1) It is unclear how these people were selected for the interviews. Moreover, interviews based on a mere 12 respondents raises questions regarding how representative this sample is of all those who are working with female migrant workers, unaccounted or not, and their children. (2) There are also concerns regarding the "semi-structured interviews." Given the range of organizations involved were any of the questions asked the same for all the respondents or did they vary? If some of the questions were standardized then what were these questions? Since the research was conducted with human participants was ethics approval sought and was it granted to proceed with this qualitative study of government social workers and administrators and staff at a Mosque?

Response 2: (1) We have added disucssion about the sampling approach that this study adopted purposive sampling in order to comprehensively capture the precarious status of UMWs and their children as well as examine the accessibility and availability to current social services from professional perspective (line 226-238). Until now, because Taiwan central government hestitates to deal with issues regarding the unaccounted-for migrant workers and their undocumented children and didn’t have any concrete policies and clear direction to solve the complex precariousness . Professionals in this specific city have developed practices dealing with related issues. Thus, although the sample size of participants seem limited, they are the most representative sample for the reseach inquires.

(2) We have enhaced the discussion about the interview protocal and analytical methods. (line 239-252). Semi-structured interviews were based on interview protocols including questions on complex issues, mainly regarding MWs’ reasons for leaving legal employment and their lives, services developed for UMW and undocumented children, and challenges and strategies of addressing the precarious status. We also added description that how the interviews were conducted and how data was analyzed (line 252-264)

Point 3: What also appears to be missing from this study is the acknowledgement of the inherent biases of the sample of informants whether they are professional biases of State officials or the disciplinary biases of social workers or administrators. This could impact the study in various ways, particularly, given the small size of the sample involved.

Related to this is the question of the relative size of the population of concern, that is, the estimated number of unaccounted female migrant workers with children. I do not recall whether any statistics were presented in this regard. While the different patterns were outlined with respect to those who choose to abandon their children or to return to their countries of origin, etc., I do not recall whether any estimates were provided of the numbers in these various pathways of the mothers and their children. Admittedly, this might be extremely difficult to estimate properly but overall this might help to provide a sense of the most prevalent ways in which the mothers and their children respond to their circumstances. This would have an obvious bearing on how one might wish to respond to these concerns through the appropriate public policy meansures. And, related to this is how many unaccounted female migrant workers are there without children, that is, never had a pregnancy?

Response 3: Thanks for reviewer’s recommendation. This article is our first step to explore the barriers to health and social services for unaccounted-for female migrant workers and their undocumented children with precarious status in Taiwan from multiple systems, particullarly including local government, child welfare placement, migrant worker detention center, hospital, regional religious center, and a foreign country office. We had do our best to recruit all the possible/potential stakeholders who are representative and familiar with providing related services and care for those UMWs and their undocumented children. Abosultely, subjective views of the UMWs themselves are absent in this research. We had already mentioned as our limitations in the part of 4.4. Limitations and implications for future research (line 651-660).

The inquiry of the the estimated number of unaccounted female migrant workers with children and without children as well as any estimates of the numbers in these various pathways of the mothers and their children are really full of difficultities to speculate. We hope that we can attain this goal when we can reach more unaccounted-for female migrant workers in the near future.

Point 4: With respect to the anaylsis of the interview data gathered it was unclear how the quotes that were selected were derived. There was some mention of continually discussing themes that emerged from the interviews but it was unclear why these specific quotations were selected. The quotations from the interviews were in italics but are these direct quotations or are they some refined quotation based on a compilation of several interviews? Direct quotations should be in quotation marks and the sources need to be specified such as (social worker, interviewed on June 21, 2020). For some quotations there was an effort at identifying the source but not at others. How long were the interviews on average? How were the audio recordings and transcriptions of the interviews stored securely, along with the consent forms of the respondents?  

I found the analysis of the interview data quite limited. There is no indication of whether there were the same or similar responses from those interviewed on any standardized questions. Was there any noticeable variance among those interviewed? One might assume that administrators might have different views than social workers, as an example, given the issues under consideration. Likewise for the staff who were interviewed at the Mosque.

Response 4: We really appreciate the reviewer’s wonderful suggestions and we had clearly revise the quotation and the sources already are specified.

We also added sentences to decribe how data was analyzed as line 258-264.

Point 5: A further comment on the findings. My sense was that there was little engagement here with the literature on this topic and/or with government policies, whether local or national. The suggestion that further research with female migrant workers with children, unaccounted or not, is, I think, essential before offering any concrete policy recommendations on this subject. And an obvious limitation of this research study is the small size of the sample. It would be hazardous to offer any definitive recommendations based on these interviews alone.

Response 5: Services for unaccounted-for migrant workers (UMWs) and their undocumented childern was involved different policies, including labor, social welfare, policing, and migration policies as well as was intersectionalized with different official sectors, even national and local government. Until now, regardless of central and local government, lack of clear policy and a main responsible ministry coordinate the complex issue. We have mentioned this concern in line 226-234 and indicated that although the sample size is small, these participants are the most representative on this topic.

We also added discussions about the small sample size as part of the study limitations (line 652-661).

Point 6: I also found the following errors in the text:

Line 92        Should it not be "substantially."

Line 127      It is UMWs and not UNWs. 

Line 165      There is an "r" missing in "counties" that should be "countries."

Line 309      Should it not be "concerns."

Line 342      A word seems to be missing. Is it not, "This decision "could" harm ..."

Lines 385 and 386   Should there not be a citation here when defining the term "permanency"?

Response 6: Thanks the reviewer for reading our article so carefully to improve the quality. We had revised the typo as the following.

Line 93      replace “substantial” to "substantially."

Line 128     replace “UNWs” to “UMWs”. 

Line 165     replace “counties” to “countries”.

Line 335    replace “concern” to “concerns”.

Line 369    Add “could” become “This decision could harm ... ”

Lines 412 and 413   Add a citation when defining the term "permanency"?

Response to Reviewer 2 Comments

Point 1: This is an interesting topic but I’m not sure that it should be published in the existing form.  The underlying research isn’t really substantial enough to support the analysis and conclusions that are offered.  Twelve cases is insufficient for this type of study.  While sample size is often controversial in qualitative research, most researchers would argue that this is far too low (Think in terms of 25-30). 

You also need to provide more information about how you analyzed the data. There is a lot of good material in the early parts of the article.  I’d consider taking the research part out, writing it up as an issues paper and then use the research (with the additional cases) for a second article.

Response 1: We really appreciate the reviewer’s delightful recommendations which remind us to increase the number of cases to provide more materials/context to explain the phenomeon of disaccounted-for female migrant workers (UMWs) and their undocumented children.

This article is our first step to explore the barriers to health and social services for unaccounted-for female migrant workers and their undocumented children with precarious status in Taiwan from multiple systems. We had do our best to recruit all the possible/potential stakeholders who are representative and the most familiar with providing related services and care for those UMWs and their undocumented children. As indicated in line 226-234. Abosultely, subjective views of the UMWs themselves are absent in this research. We had already mentioned as our limitations in the part of 4.4. Limitations and implications for future research (in line 652-661).

As the reviewer 02’s suggestion, we also added discussions about the small sample size as part of the study limitations (line 652-661).

We do appreciate the review’s recommendcation on focusing on an issue paper. But we consider that we do need to provide some evidence to raise the concerns of the current issue and trend and to welcome more professionals to join us advocating for the vulnerable population. Our second article/next step will be on the topic about the shelter with undocument children and try to reach their run-away/hiden female migrant workers as mothers (UMWs) to prolonged engage and persistantly observe as well as interview in order to capture the dynamic process and subjective interpretation of their current living and precarious condition. We also will try to increase the sample size to 25-30 cases through the snowball sampling to improve the credibility of our research. Again, we really appreciate the reviewer’s thoughtful suggestions.

We also added sentences to decribe how data was analyzed as line 257-264.

Reviewer 2 Report

This is an interesting topic but I’m not sure that it should be published in the existing form. 

The underlying research isn’t really substantial enough to support the analysis and conclusions that are offered.  Twelve cases is insufficient for this type of study.  While sample size is often controversial in qualitative research, most researchers would argue that this is far too low (Think in terms of 25-30).  You also need to provide more information about how you analyzed the data.

There is a lot of good material in the early parts of the article.  I’d consider taking the research part out, writing it up as an issues paper and then use the research (with the additional cases) for a second article.

Author Response

Response to Reviewer 2 Comments

Point 1: This is an interesting topic but I’m not sure that it should be published in the existing form.  The underlying research isn’t really substantial enough to support the analysis and conclusions that are offered.  Twelve cases is insufficient for this type of study.  While sample size is often controversial in qualitative research, most researchers would argue that this is far too low (Think in terms of 25-30). 

You also need to provide more information about how you analyzed the data. There is a lot of good material in the early parts of the article.  I’d consider taking the research part out, writing it up as an issues paper and then use the research (with the additional cases) for a second article.

Response 1: We really appreciate the reviewer’s delightful recommendations which remind us to increase the number of cases to provide more materials/context to explain the phenomeon of disaccounted-for female migrant workers (UMWs) and their undocumented children.

This article is our first step to explore the barriers to health and social services for unaccounted-for female migrant workers and their undocumented children with precarious status in Taiwan from multiple systems. We had do our best to recruit all the possible/potential stakeholders who are representative and the most familiar with providing related services and care for those UMWs and their undocumented children. As indicated in line 226-234. Abosultely, subjective views of the UMWs themselves are absent in this research. We had already mentioned as our limitations in the part of 4.4. Limitations and implications for future research (in line 652-661).

As the reviewer 02’s suggestion, we also added discussions about the small sample size as part of the study limitations (line 652-661).

We do appreciate the review’s recommendcation on focusing on an issue paper. But we consider that we do need to provide some evidence to raise the concerns of the current issue and trend and to welcome more professionals to join us advocating for the vulnerable population. Our second article/next step will be on the topic about the shelter with undocument children and try to reach their run-away/hiden female migrant workers as mothers (UMWs) to prolonged engage and persistantly observe as well as interview in order to capture the dynamic process and subjective interpretation of their current living and precarious condition. We also will try to increase the sample size to 25-30 cases through the snowball sampling to improve the credibility of our research. Again, we really appreciate the reviewer’s thoughtful suggestions.

We also added sentences to decribe how data was analyzed as line 257-264.

Round 2

Reviewer 1 Report

This is substantial improvement over the original submission. Nonetheless, I would be inclined to indicate and emphasize that the qualitative interview information collected is presented merely for illustrative purposes and that it is suggestive and in no way is intended to be representative of the population of the services workers in Taiwanese society who work with UMWs and their children.

The text still contains numerous writing errors that should be cleaned up and correction. In addition, the text ought to be thoroughly vetted before publication. What follows are merely examples of other errors that ought to be corrected. In short, the submission needs a careful proofreading of the text. For instance,

Line 51  -  It should be "the Taiwanese ...." "'The' is missing."

Line 58  -  An "a" is missing. "... representing "a" 220 increase ... "

Line 61  - It should be "for illegal" and not "to illegal."

     227  -  "... there has been no ... " The word "been" is missing.

     503  -  It is "prepare" and not "prepares."

     540  -  Not "forcedly" but "forcibly."

     691  - "... respond to this issue ..." The "to" is missing.

Reviewer 2 Report

Thank you for your revisions.  While the revisions do make the paper stronger, they do not change the fundamental problem that you have an insufficient number of cases to support your analysis and conclusions. Your statement that "We had do our best to recruit all the possible/potential stakeholders who are representative and the most familiar with providing related services and care for those UMWs and their undocumented children" may be true, but there are scientific standards for this type of research that your work does not meet. Your statement " But we consider that we do need to provide some evidence to raise the concerns of the current issue and trend and to welcome more professionals to join us advocating for the vulnerable population" documents why this is an issue.  Evidence that does not meet minimum standards should never be used for decision-making, especially with vulnerable populations. While you do mention this in the limitations, I cannot see why you would want to provide this data at this point. Is that really ethical?

In good conscience, I cannot recommend this for publication.